# Microbial Community Structure in the Sediments and Its Relation to Environmental Factors in Eutrophicated Sancha Lake

**DOI:** 10.3390/ijerph16111931

**Published:** 2019-05-31

**Authors:** Yong Li, Jiejie Zhang, Jianqiang Zhang, Wenlai Xu, Zishen Mou

**Affiliations:** 1Faculty of Geosciences and Environmental Engineering, Southwest Jiaotong University, Chengdu 610059, China; liyong@swjtu.edu.cn (Y.L.); jiejiezhang90@my.swjtu.edu.cn (J.Z.); 2State Key Laboratory of Geohazard Prevention and Geoenvironment Protection, Chengdu University of Technology, Chengdu 610059, China; mouzishen17@cdut.edu.cn; 3Haitian Water Grp Co Ltd., Chengdu 610059, China

**Keywords:** Sancha Lake sediments, eutrophication, high-throughput sequencing, microbial community, environmental factors

## Abstract

To study the microbial community structure in sediments and its relation to eutrophication environment factors, the sediments and the overlying water of Sancha Lake were collected in the four seasons. MiSeq high-throughput sequencing was conducted for the V3–V4 hypervariable regions of the 16S rRNA gene and was used to analyze the microbial community structure in sediments. Pearson correlation and redundancy analysis (RDA) were conducted to determine the relation between microbial populations and eutrophic factors. The results demonstrated four main patterns: (1) in the 36 samples that were collected, the classification annotation suggested 64 phyla, 259 classes, 476 orders, 759 families, and 9325 OTUs; (2) The diversity indices were ordered according to their values as with summer > winter > autumn > spring; (3) The microbial populations in the four seasons belonged to two distinct characteristic groups; (4) pH, dissolved oxygen (DO), total phosphorus (TP), and total nitrogen (TN) had significant effects on the community composition and structure, which further affected the dissolved total phosphorus (DTP) significantly. The present study demonstrates that the microbial communities in Sancha Lake sediments are highly diverse, their compositions and distributions are significantly different between spring and non-spring, and Actinobacteria and Cyanobacteria may be the key populations or indicator organisms for eutrophication.

## 1. Introduction

Eutrophication has become a worldwide environmental problem. Sediments are a source of endogenous nutrients in eutrophic waters and have a significant impact on eutrophication [1]. As the primary driving force of natural material circulation, microbes in the sediments play an essential role in material circulation between water and sediments as well as in eutrophication [2,3]. In eutrophic waters, microbes in the sediments are the most important contributors to the conversion of complex organic matter (OM) and mineral elements [4], and the studies on their diversity and structure are particularly important for understanding the eutrophic aquatic ecosystems [5]. Exploration of the main ecological and environmental factors affecting the microbial community composition has been one of the focuses of relevant scholars [6].

Jin et al. [7] used terminal restriction fragment length polymorphism (T-RFLP) and Biolog-Eco microplates to study the microbial diversity and structure of Poyang Lake sediments. The results showed that there was synergistic spatial heterogeneity in microbial community structure and diversity and the heavy metal Zn had a significant impact on microbial community structure and diversity. Fan et al. [8] conducted high-throughput sequencing to study the diversity of microbial communities in sediments of Taihu Lake in winter. It was found that Nitrospirae and Acidobacteria were the dominant phyla, and nutrient levels, water temperature, and pH could affect the microbial community structure of the sediments. Yang et al. [9] used high-throughput sequencing technology to study the characteristic differences in microbial community structure of sediments from various parts of Nanfei River. They found that Proteobacteria, Chloroflexi, and Bacteroidetes were the dominant phyla and that Proteobacteria and Bacteroidetes were affected more by sediments total phosphate (TP), while Chloroflexi were more affected by weak acid extractable form nitrogen (WAEF-N) in sediments. Liu et al. [10] also employed high-throughput sequencing to analyze the composition and spatial distribution of microbial communities in the sediments of Bohai Bay. They revealed that Proteobacteria, Acidobacteria, and Chloroflexi were the dominant populations, and the particle size and organic nitrogen content of the sediments had a significant effect on the species composition and community structure. These studies were basically about shallow lakes, rivers, and bays and focused on the effects of environmental physicochemical factors on microbial populations during eutrophication.

There are few studies on the diversity and structure of microbial communities in the eutrophic sub-deep reservoirs and even less on the effects of microbial communities on eutrophication. SAI·Bayartu et al. [11] used the epifluorescent microscopic direct counting method (FDC) to study the spatial distribution of bacterial abundance in Lake Bosten sediments and its relationship with environmental factors. They found that the bacterial abundance was significantly positively correlated with permanganate index (COD_Mn_) and Cl^−^ and also revealed that the bacterial spatial distribution was a sensitive indicator of eutrophication.

Peng et al. [1] used PCR-denaturing gradient gel electrophoresis (PCR-DGGE) to study microbial diversity and seasonal changes in sediments. The results showed that Proteobacteria and Bacteroidetes were the dominant populations, and temperature was the key factor affecting the diversity of bacterial community structure in sediments. Qu et al. [12] studied the longitudinal distribution of bacterial diversity in the sediments of Guanting Reservoir by amplified ribosomal DNA restriction analysis (ARDRA) of the 16S rDNA in a cloned library. They found that the dominant bacteria were β or γ-*Proteus bacilli*. However, those studies on bacterial community diversity in sub-deep reservoirs only employed traditional methods such as FDC, PCR-DGGE, and ARDRA. With the development of sequencing in molecular biology, high-throughput sequencing technology has been proven to be an efficient tool for identifying the entire profile of microbial communities and has been widely applied in research of microbial community structure in complex environments [13]. By measuring environmental physicochemical factors, DNA extraction, PCR amplification, and high-throughput sequencing, this present study analyzed the diversity, structure, and spatiotemporal distribution of bacterial communities in Sancha Lake sediments and their relation to eutrophic factors such as dissolved oxygen (DO), TP, and total phosphorus (DTP). We attempted to explore the main environmental factors affecting the microbial community structure and revealed the microbial populations exerting great impacts on the eutrophication of Sancha Lake. We assumed that the microbial community composition of the sediments in Sancha Lake is rich and diverse and the structure and composition of the microbial community are closely related to eutrophication.

## 2. Materials and Methods

### 2.1. Site Description and Sample Collection

Sancha Lake is located in Tianfu New District, Sichuan Province, China, 30°13′08″–30°19′56″ N, and 104°11′16″–104°17′16″ E. It is an important source of drinking water and has serious eutrophication issues. The hydrological and morphological characteristics of Sancha Lake and the description of the sampling sites of this study have been described in detail in previous studies [14]. At the nine chosen sampling sites shown in Figure 1, the surface sediments of the lake bed were collected with a Peterson grab and then sealed in polyethylene bags in April 2017 (spring), August 2017 (summer), November 2017 (autumn), and January 2018 (winter). Three parallel samples were collected for each sampling site and then mixed as the representative sample of that site. After the samples were transported to the laboratory, part of each sample was stored at 4 °C for physical and chemical analyses (within 24 h), and another part of each sample was stored at −80 °C for DNA extraction [15]. At the same time, at each sampling site, a gas-tight water sampler was used to collect the overlying water on the sediments surface for the analysis of the water environment index [15].

### 2.2. Determination of Physicochemical Factors of the Sediments and Overlying Water

TP, inorganic phosphorus (IP), organic phosphorus (OP), phosphonium hydroxide (NaOH-P), phosphorous hydrochloride (HCl-P), total organic carbon (TOC), total nitrogen (TN), DTP, pH, and temperature (T) were measured as previously described by Li et al. [14].

### 2.3. DNA Extraction and PCR Amplification

Total DNA was extracted according to the instructions of the Power Water^®^ DNA Isolation Kit (MO BIO laboratories, Inc., Carlsbad, CA, USA). The concentration and purity of the DNA extracts were measured with a NanoDrop2000 (Thermo Fisher Scientific, Waltham, MA, USA), and the quality of the DNA extracts was determined by 1% agarose gel electrophoresis. PCR amplification of the V3–V4 hypervariable regions of the 16S rRNA gene was performed using 338F(5’-ACTCCTACGGGAGGCAGCAG-3’) and 806R (5’-GGACTACHVGGGTWTCTAAT-3’) primers [16]. The PCR protocol was as follows: initial denaturation at 98 °C for 2 min, 27 cycles of (denaturation at 98 °C for 15 s, annealing at 55 °C for 30 s, extension at 72 °C for 30 s), and finally extension at 72 °C for 5 min. Each sample involved three technical replicates for 16S rRNA gene amplification, which was carried out in a GeneAmp 2720 thermocycler (Applied Biosystems, Foster City, CA, USA). Indexing PCR was performed in a 25 μL reaction system containing 5 μL reaction buffer (5×), 5 μL GC buffer (5×), 2 μL 2.5 mM dNTP, 1 μL Forward primer (10 μM), 1 μL Reverse primer (10 μM), 0.25 μL Q5^®^ High-Fidelity DNA Polymerase, 2 μL template DNA, and 8.75 μL ddH2O.

### 2.4. Illumina Miseq Sequencing and Data Processing and Analysis

The PCR product was recovered from a 2% agarose gel, purified with an AxyPrep DNA Gel Extraction Kit (Promega, Madison, WI, USA), eluted with Tris-HCl, detected by 2% agarose gel electrophoresis, and quantified by a QuantiFluorTM-ST Fluorometer Instrument (Promega, Madison, WI, USA). The purified amplicons were used to construct a library of PE 300 according to the standard operating procedure of the Illumina MiSeq platform (Illumina, San Diego, CA, USA).

The sequencing was performed on an Illumina’s Miseq PE300 platform (provided by Shanghai Majorbio Bio-Pharm Technology Co. Ltd., Shanghai, China).

The control of the original sequences was conducted using the Trimmomatic software, and the sequences were merged with the FLASH software (version 2.7, http://ccb.jhu.edu/software/FLASH/, Center for Bioinformatics and Computational Biology, Iowa City, IA, USA). The operational taxonomic unit (OTU) clustering of sequences was performed with the UPARSE software (version 7.1 http://drive5.com/uparse/, Edgar, RC, Tiburon, CA, USA) based on 97% similarity, with all singletons and chimeras removed concurrently. The Ribosomal Database Project (RDP) classifier was used to taxonomically annotate the most abundant sequence in each OTU, using the SLIVA database (Release115, http://www.Arb-SLIVA.de) [17] as a reference and with an alignment quality threshold of 70%. On the basis of the taxonomic information for each OTU, R was used for statistical analysis and plotting of the community structure.

On the basis of the OTU clustering results, the QIIME software (version 2.0, http://qiime.org/, Rob Knight Lab, Boulder, CO, USA) was used to perform species richness analysis (e.g., abundance-based coverage estimators (ACE) and Chao1 indices) and diversity analysis (Shannon and Simpson indices) for each sample [18].

To compare the difference in taxonomic composition between sample groups, the Metastats algorithm (http://metastats.cbcb.umd.edu/) from the Mothur software (Version 1.35.1, Patrick Schloss, University of Michigan, MI, USA) was used to analyze the difference of sequence number of each taxon between groups [19]. The difference in quantity was tested by pairwise comparison. According to the test results, R was used to plot the richness distribution map for the most significantly different taxa between groups.

To investigate the similarity of community structure among samples, the non-metric multidimensional scaling (NMDS) [20] was adopted. The QIIME and R were used to naturally decompose the community data structure. By sorting the samples, temporal and spatial differences in the structure and composition of microbial communities in the samples were observed and clarified.

### 2.5. Statistical Analysis

Statistical analysis was performed using the statistical product and service solutions (SPSS) statistical software (version 20.0, IBM, Armonk, NY, USA). One-way analysis of variance (ANOVA) was used to test differences between seasons. The Pearson correlation coefficient was used to investigate the correlation between environmental factors and microbial populations in sediments. The redundancy analysis (RDA) was performed on the microbial communities and environmental factors in the sediments using the vegan package for R [8]. Statistical significance was determined at the confidence levels of 0.05.

## 3. Results and Discussion

### 3.1. Physicochemical Properties of the Sediments and Overlying Water

The physicochemical properties of sediments and overlying water are shown in Table 1. The pH of the overlying water ranged from 6.06 to 7.73, was weakly acidic in spring, weakly alkaline in summer, and neutral in autumn and winter. The temperature of the overlying water ranged from 11.1 to 20.0 °C, with the highest value in summer and the lowest value in spring and winter. The value of DO ranged from 4.10 to 9.40 mg·L^−1^, was the highest in spring, lower in summer, and the lowest in autumn and winter. The value of DTP ranged from 4.10 to 9.90 mg·L^−1^, being the highest in spring, lower in autumn and winter, and the lowest in summer. The value of TOC ranged from 5.7 to 30.3 mg·L^−1^, being higher in summer than in spring, and neutral in autumn and winter. The value of TN ranged from 1.2 to 3.1 mg·L^−1^, with the highest values in summer, the lowest in winter, and neutral values in spring and autumn. 

The TP content in the sediments ranged from 291 to 4562 mg·kg^−1^, and the average content was higher than those of Poyang Lake sediments (689.34 mg·kg^−1^) [7], Yaohu Lake sediments (987.93 mg·kg^−1^) [21], and Guanting Reservoir sediments (1268.93 μg·g^−1^) [12]; also, it was at the high end when compared to those of lakes of the same type. The TP content varied dramatically across various sampling sites and changed hugely seasonally for the same sampling site. Basically, all sampling sites exhibited a trend of high TP content in summer and autumn and low TP content in winter and spring, consistent with the report by Xie et al. [22]. The sampling sites L8, L6, L5, and L9 had high sediment TP contents, with L8 and L9 being the highest. Historically, those sampling sites were high-concentration areas for cage fish culture, thus the high TP contents may be caused by excessive fish feed accumulated on the lake bottom. The site L9 is adjacent to an area with high human activity, and the high TP contents may be due to the deposition of OM discharged from the wastewater. The sediments TP contents of L4 and L1 sites were relatively high too. These areas were historically the concentrated areas of cage fish culture, and the relative high TP contents may also be caused by the accumulation of fish feed on the lake bottom. Because of the large flow of water and less sedimentation of input pollutants, the TP contents of the headwater area L4 and the dam tailwater L2 and L3 sites were lower than those of the other sampling sites. According to the “Nutrient Criteria Technical Guidance Manual for Lakes and Reservoirs” [23], the sediments TP levels suggested heavily pollution (>650 mg·kg^−1^).

The TP in the sediments consisted of IP and OP, the former of which accounted for about 70% of the TP content and played a dominant role in the trend of TP. There were two occurrence forms for IP, primarily, NaOH-P and HCl-P. The spatial and temporal distributions of the various forms of phosphorus, such as IP, OP, NaOH-P, and HCl-P, were consistent with those of TP. 

### 3.2. Analysis of Diversity and Richness of Microbial Communities

A total of 36 samples were collected from the 9 sample sites of Sancha Lake in the four seasons. High-throughput sequencing of 16S rRNA was performed on an Illumina Miseq. The obtained data were subjected to quality control, splicing, and chimera removal, and resulted in a total of 1,370,626 valid sequences, with an average length of 418.747 bp. The sequence numbers of samples ranged from 26,738 to 61,872, with an average of 38,073. The 97% similarity criterion was used for the microbial OTU clustering of each sample. In the 1,370,626 sequences, a total of 9325 OTUs were obtained. The sediments’ OTU comparison of the four seasons is displayed in the Venn map (Figure 2). There were 5559 OTUs in the four seasons altogether, with 945 unique OTUs in spring, 245 unique ones in summer, 211 in autumn, and 350 in winter. The OTU number was ordered as summer > winter > autumn > spring, and the effective sequence number was ordered as spring > summer > winter > autumn. The OTU number of a single sample ranged from 730 to 2785. 

The microbial diversity indices (Shannon and Simpson) and richness indices (ACE and Chao1) were calculated according to the OTU results of the abovementioned sediments. The Shannon index of each sample was between 5.02 and 10.22. The Shannon index of the reservoir tailwater L3 site in summer was the highest, while that of the L9 site adjacent to the human activities area in spring was the lowest. The Shannon index could be sorted as summer > winter > autumn > spring. The Simpson index of each sample was between 0.8535 and 0.9981, with the highest value for the concentrated original cage culture L6 site in summer and the lowest one for the human activities-concentrated site L9 in spring, with summer > winter > autumn > spring as well. The ACE index of each sample was between 518.65 and 3422.62, and the chao1 index was between 499.17 and 3173.02. The trends of richness indices and diversity indices were similar considering sampling sites and seasons.

Qu et al. [12] obtained a total of 309 positive clones and 153 genotypes from sediments by ARDRA, demonstrating a low diversity index. This present study analyzed the microbial diversity and community structure of Sancha Lake sediment samples by Illumina Miseq system and obtained a high diversity index. The microbial diversity and community composition of sediments could be accurately analyzed. Similar to the results of Peng et al. [1], there were spatial and temporal differences in the microbial population diversity in Sancha Lake regarding different regions and seasons, and the seasonal variations were more significant.

### 3.3. Analysis of the Taxonomic Composition and Spatiotemporal Changes of Microbial Communities

The RDP classifier was used to taxonomically annotate the most abundant sequence in each OTU. The taxonomic information of each OTU was obtained by SLIVA alignment. A total of 64 phyla, 259 classes, 476 orders, 759 families, and 1310 genera were detected in the 36 sediment samples from the 9 sampling sites in Sancha Lake. The microbial communities with a relative species abundance of more than 1% involved 30 phyla, 59 classes, 96 orders, 127 families, and 148 genera. The distribution and relative species abundances of the top 20 abundant microbial populations in the sediments of all sampling sites in the four seasons are shown at the phylum level in Figure 3. Microbial populations with higher absolute abundance were ordered as follows: Proteobacteria (accounting for 34.5%, sequence number 177,148), Actinobacteria (12.1%, 63,248), Chloroflexi (10.8%, 54,834), Bacteroidetes (9.7%, 49,488), Firmicutes (4.3%, 21,688), Cyanobacteria (3.9%, 20,421), Verrucomicrobia (3.7%, 18,582), Acidobacteria (3.3%, 17,322), Planctomycetes (1.4%, 6964), Spirochaete (2%, 9914), Nitrospirae (2%, 9924), Aminicenantes (1.3%, 6570), and Ignavibacteriae (1.2%, 5883). Proteobacteria and Actinobacteria were dominant microbes in the sediments of Sancha Lake. Proteobacteria mainly included families such as Syntrophaceae (3.9, 19,332), Rhodocyclaceae (2.2%, 10,905), Comamonadaceae (2.0%, 9914), Hydrogenophilaceae (1.6%, 7,931), Xanthomonadales Incertae Sedis (1.3%, 6570), and Helicobacteraceae (1.0%, 4957). Actinobacteria mainly included families such as Sporichthyaceae (8.5%, 42,135) and Acidimicrobiaceae (1.4%, 6964).

With respect to the spatial distribution of the microbial communities, although the relative abundances of different microbial population varied, basically the abundant bacteria were present at all sites. The highly concentrated original cage culture site L6 in summer had the highest number of microbial populations, and these microbial populations covered 64 phyla, while the site L9 adjacent to the human activities area in spring had the lowest number of microbial populations, only involving 14 phyla. The microbial populations (relative species abundance >1%) detected in all nine sampling sites of Sancha Lake were Proteobacteria (relative species abundance 20.20–50.43%), Actinobacteria (0.95–62.21%), Bacteroidetes (1.42–22.93%), Firmicutes (0.18–19.55%), Cyanobacteria (0.38–24.22%), Verrucomicrobia (0.41–8.56%), Chloroflexi (0.24–30.58%), Acidobacteria (0.06–10.04%), and Planctomycetes (0.10–4.0%). The Proteobacteria and Actinobacteria had the highest relative abundance and were widely distributed across all sites in all seasons with absolute superiority. Among the other populations, the Nitrospirae (0–6.62%) were not distributed at the highly concentrated original cage culture site L5 or the intensive-human-activity adjacent site L9 in spring and was primarily distributed in the tailwater sites L1 and L2; the Spirochaetae (0–8.50%) were not found at the L3, L7, and L9 sites in spring and were primarily distributed at the deepwater sites L4 and L6 sites in summer and autumn, while the Aminicenantes (0–3.41%) and Ignavibacteriae (0–3.12%) were not detected at any site in spring but distributed over all sites in summer, autumn, and winter.

Regarding the temporal distribution of the microbial communities, it was significantly different with respect to the season, and seasonal changes had a large impact on the microbial structure. Microbial species in the sediments were the most abundant in summer and the least abundant in spring. The most abundant microbial populations were present in all four sampling seasons, but with significantly different abundances. The microbial populations (relative species abundance >1%) detected in all seasons were Proteobacteria, Actinobacteria, Bacteroidetes, Firmicutes, Cyanobacteria, Verrucomicrobia, Chloroflexi, Acidobacteria, Planctomycetes, Spirochaetae, and Nitrospirae. From the Metastats test results and the violin plot combined with the box plot (Figure 4), it can be seen that at the phylum level, the absolute abundances (sequence number) of the six phyla (Actinobacteria, Cyanobacteria, Acidobacteria, Aminicenantes, Chloroflexi, and Ignavibacteriae) were significantly different (*p* < 0.01) between spring and the non-spring seasons. The phyla Actinobacteria and Cyanobacteria had the largest absolute abundances in spring and very small abundances in the non-spring seasons, whereas Acidobacteria, Aminicenantes, Chloroflexi, and Ignavibacteriae behaved the opposite way. In spring, the succession of microbial community structure established the dominant position of Actinobacteria and Cyanobacteria, which in turn seriously affected the spring bloom of Sancha Lake. Therefore, they might be the key populations or indicator organisms in eutrophication.

At the genus level, the distribution and relative abundances of the top 20 abundant microbial populations in the sediments of Sancha Lake in the four seasons (spring, summer, autumn, and winter) are shown in Figure 5, and the bottom 20 microbial groups and uncultured bacteria are grouped together as “others”. The genera with high absolute abundance (>1%) were hgcI_clade (accounting for 8.4%, sequence number 41,639), Candidatus_Xiphinematobacter (1.6%, 7931), Dechloromonas (1.5%, 7436), Smithella (1.3%, 6444), Syntrophus (1.3%, 6444), Sulfuricurvum (1.1%, 5453), and Thiobacillus (1.0%, 4957). The genus hgcI clade was the most abundant one, accounting for 69.4% of the Actinobacteria phylum. It was the dominant population in the sediments of Sancha Lake and was mostly detectable in spring. The other genera with high absolute abundances (>1.0%) were distributed across all seasons.

The bacteria classified at the genus level included a large number of unclassified and uncultured bacteria (36.2%), bringing difficulties to the study of the microbial community structure. Some uncultivated bacteria at the genus level in the Anaerolineaceae family (Anaerolineaceae _uncultured) had an absolute abundance of 4.2%, accounting for 38.9% of the Chloroflexi phylum, and were primarily distributed in summer, autumn, and winter. Some uncultivated genera in the Xanthomonadales Incertae Sedis family (Xanthomonadales Incertae Sedis_uncultured) had an absolute abundance of 1.1%, accounting for 3.1% of the Proteobacteria phylum, and were distributed in all seasons. Some uncultivated bacteria genera in the Nitrospiraceae family (Nitrospiraceae_uncultured) had an absolute abundance of 1.0%, accounting for 50% of the Nitrospirae phylum, and were mainly distributed in summer, autumn, and winter.

A total of 64 phyla were obtained in this study. Proteobacteria were found to be the dominant population in the sediments of all four seasons and nine sampling sites. Yang et al. [9] also obtained a similar result when studying the sediments of different parts of Nanfei River, demonstrating that Proteobacteria had obvious advantages in 17 sediment sampling sites, accounting for 54% of the total sequences. Similarly, in their study about the composition and spatial distribution of microbial community in the sediments of the Bohai Bay mouth, Liu et al. [10] revealed Proteobacteria as the primary dominant population in 21 sediment samples, accounting for 56.8% of the whole communities. Peng et al. [1] reached a similar conclusion in the study of microbial community diversity in the sediments of East Lake by PCR-DGGE. Zhang et al. [24] used the 16S rRNA gene clone library to study the microbial community diversity in the sediments of the smallmouth lake area of Ulansuhai Nur and concluded similarly. However, the results of the present study are yet different from those reported, primarily in relation to the different populations. In spring, the microbial community structure succession established the dominant positions of Actinobacteria and Cyanobacteria, exerting an important influence on the spring bloom of Sancha Lake, and presumably representing the key microbes or indicator organisms for eutrophication. Ye et al. [25] demonstrated similar results in the study of the distribution characteristics of archaea and bacteria in the sediments of eutrophic Taihu Lake. At the genus level, the genus hgcI_clade was often found to be the dominant population in the water of the rivers, lakes, and reservoirs but was seldom found in sediments [26]. However, the results of the present study showed that the genus hgcI_clade was the dominant population in the sediments of Sancha Lake. Keshri et al. [27] reached a similar conclusion in the study of microbial community diversity in the sediments of freshwater lakes in all four seasons.

### 3.4. Analysis of Microbial Community Structure Differences

The distance matrix of weighted UniFrac was calculated with QIIME, and R was used to plot the NMDS analysis to further evaluate the similarities and differences in the microbial community composition in different samples of Sancha Lake sediments. From the NMDS map (Figure 6), it can be seen that the microbial communities of the nine sampling sites in spring were concentrated in the third quadrant. The microbial communities of the 27 samples from the 9 sampling sites in summer, autumn and winter were scattered in the second and third quadrants, all on the right of the vertical axis, exhibiting far distances from the area with microbial community of the 9 sampling sites in spring. The microbial community structure in spring was very different from those of the other three seasons, so there were significantly seasonal differences between spring and non-spring. The microbial community structures in summer, autumn, and winter were different (scattered in the ranking diagram), indicating a possible, but not significant, impact of non-spring seasonal changes on the microbial community structure. Spatially, in spring, the microbial community structures of various samples shared a high similarity, indicating a limited impact of the sampling site type on the microbial community structure. The samples from different environments in summer, autumn, and winter were scattered in the ranking diagram, indicating an impact of the sampling site in non-spring seasons on the microbial community structure, though not significant. 

Wan et al. [28] studied the bacterial community structures of the sediments from Lake Taihu and found that they showed strong seasonal variation, with the highest abundance in spring/summer. Sun et al. [29] studied community dynamics of prokaryotic and eukaryotic microbes in an estuary reservoir and found that the bacterial community structures of estuary reservoir were classified into four groups, which were mainly composed of the summer samples, the autumn samples, the spring and winter samples from the middle-rear parts of the reservoir, and the spring and winter samples from the front-middle parts. In our study, the results showed that the irrigation activity increases the disturbance of water and DO in spring. Therefore, the microbial community structure of the sediments of Sancha Lake is primarily affected by seasonal changes between spring and non-spring.

### 3.5. Correlation Analysis of Microbial Populations and Environmental Physicochemical Factors

The RDA was performed on the relative abundance of the microbial phyla (relative species abundance >1%) and the environmental physicochemical factors of water and sediments (Figure 7). The results demonstrated that the eigenvalue on the first axis was 0.500, explaining 50.0% of the data, and the eigenvalue of the second axis was 0.413, explaining 41.3% of the data. All samples from spring were concentrated in the second quadrant, while the samples from summer, autumn, and winter were mixed and spreading in the third and fourth quadrants. According to the relationship between microbial communities at the phylum level and environmental factors, DO, pH, DTP, and TP had long lines on the first and second axes, indicating that they had large impacts, significantly correlated (*p* < 0.05), on the overall microbial community, and the DTP changes were highly significantly correlated with microbial communities in spring (*p* < 0.01).

Pearson correlation analysis was performed on the microbial populations (relative species abundance >1%) and environmental factors at the phylum level, and the correlation analysis between the two factors was shown in Table 2. The results showed that the relative abundance of Actinobacteria was negatively correlated at high significance with pH (r = −0.761, *p* < 0.01), significantly negatively correlated with TP (r = −0.344, *p* < 0.05) and HCl-P (r = −0.341, *p* < 0.05), and positively correlated at high significance with DO (r = 0.639, *p* < 0.01) and DTP (r = 0.540, *p* < 0.01). The relative abundance of Cyanobacteria was negatively correlated at high significance with pH (r = −0.674, *p* < 0.01), significantly negatively correlated with OP (r = −0.330, *p* < 0.05) and TP (r = −0.281, *p* < 0.05), positively correlated at high significance with DO (r = 0.544, *p* < 0.01), and significantly positively correlated with DTP (r = 0.33, *p* < 0.05). The relative abundance of Chloroflexi was positively correlated at high significance with pH (r = 0.685, *p* < 0.01), negatively correlated at high significance with DTP (r = −0.437, *p* < 0.01), and positively correlated at high significance with TN (r = 0.549, *p* < 0.01). Acidobacteria were positively correlated at high significance with pH (r = 0.463, *p* < 0.01) and DO (r = 0.544, *p* < 0.01) and significantly negatively correlated with DTP (r = −0.349, *p* < 0.05). Planctomycetes were positively correlated at high significance with pH (r = 0.474, *p* < 0.01), negatively correlated at high significance with DO (r = −0.438, *p* < 0.01), and significantly positively correlated with TP (r = 0.330, *p* <0.05) and HCl-P (r = 0.408, *p* < 0.05). Nitrospirae were positively correlated at high significance with pH (r = 0.330, *p* < 0.01) and significantly positively correlated with TN (r = 0.056, *p* < 0.05). Aminicenantes were positively correlated at high significance with pH (r = 0.538, *p* < 0.01), negatively correlated at high significance with DO (r = −0.466, *p* < 0.01), and significantly negatively correlated with DTP (r = −0.397, *p* < 0.05). Ignavibacteria were positively correlated at high significance with pH (r = 0.635, *p* < 0.01) and negatively correlated at high significance with DO (r = −0.438, *p* < 0.01) and DTP (r = −4.3, *p* < 0.01). 

Winters et al. [30] found that environmental factors such as nitrate, ammonium salt, and phosphate could affect the microbial community composition of sediments in the Laurentian Great Lakes. Jin et al. [7] Indicated that Zn has a significant effect on microbial community structure and diversity. Jing et al. [31] reported that ammonium salts and organic matter significantly affected the microbial community composition of sediments of the Taihu Lake. Fan et al. [8] discovered that the microbial community structure in the sediments was influenced by nutrient content, water temperature, and pH. Yang et al. [9] found that the cation exchange capacity (CEC), TOC, TP, and WAEF-N of sediments were the key factors causing differences in microbial community structure. Seymour et al. [32] reported that the microbial community structure was related with DO content. The results of the present study indicate that the microbial populations in sediments were significantly correlated with DO, pH, TP, and TN. The OP and HCl-P in different forms of phosphorus significantly affected the composition of the microbial community, and the microbial community structure had the greatest impact on DTP.

Actinobacteria and Cyanobacteria were associated at high significance with DTP. Yandigeri et al. [33] reported that Cyanobacteria were capable of dissolving inorganic phosphorus, and Streptomyces in Actinobacteria were capable of dissolving organic phosphorus. Therefore, from the results of this study, it is speculated that the changes in the overlying water DTP in Sancha Lake were partly due to the dissolution of inorganic and organic phosphorus by Actinobacteria and Cyanobacteria, respectively, releasing DTP into the overlying water. It is suggested that Actinobacteria and Cyanobacteria in sediments have an important role in eutrophication, consistent with the conclusion reported by Li et al. [14].

## 4. Conclusions

The high-throughput sequencing results showed that the 9325 OTUs from the 36 sediment samples from Sancha Lake belonged to 64 phyla, 259 classes, 476 orders, 759 families, and 1310 genera. The bacterial community structure demonstrated a high diversity. The diversity indices were ordered according to their values as summer > winter > autumn > spring. The phylum with the highest absolute abundance was Proteobacteria (34.5%), followed by Actinobacteria (12.1%) and Chloroflexi (10.8%). At the genus level, the hgcI_clade of the Actinobacteria phylum had the highest abundance and thus was the dominant population. The bacterial community structure exhibited significant differences under the influence of spring and non-spring seasons but did not vary significantly with respect to the change in sediment type. The contents of pH, DO, TP, and TN may significantly affect the dominant population (abundance >1%) at the phylum level. Actinobacteria and Cyanobacteria were significantly positively correlated with the DTP content of the overlying water, suggesting that they may be the key populations or serve as indicator organisms for the eutrophication of Sancha Lake.

## Figures and Tables

**Figure 1 ijerph-16-01931-f001:**
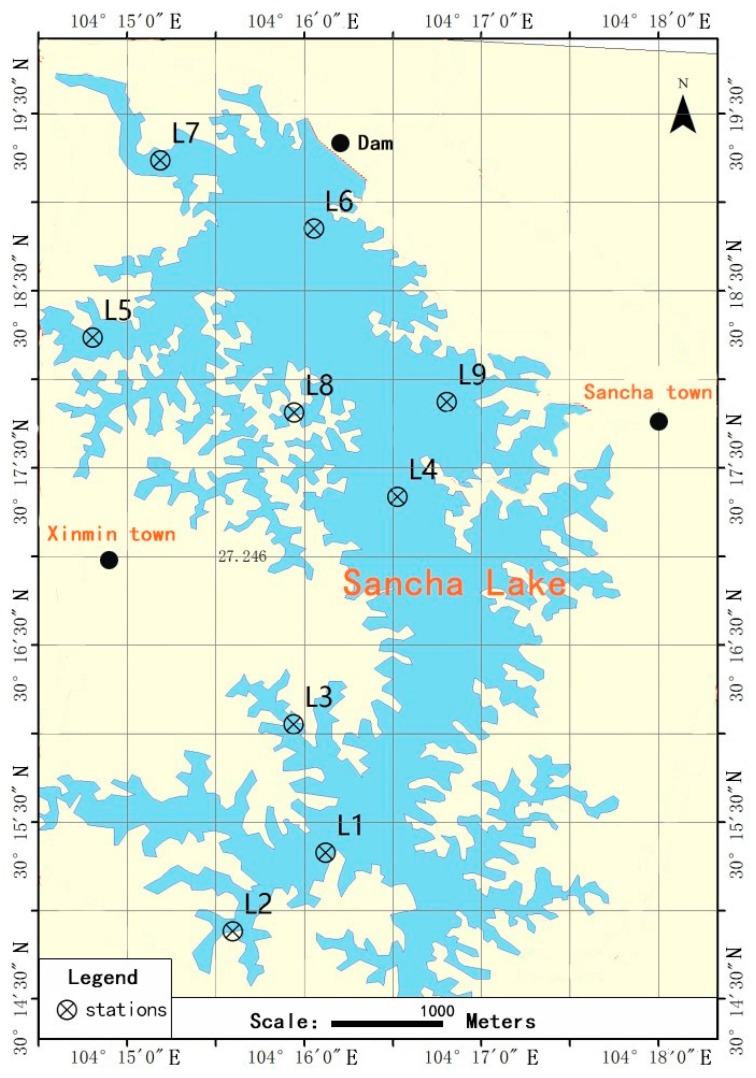
Sampling sites at Sancha Lake.

**Figure 2 ijerph-16-01931-f002:**
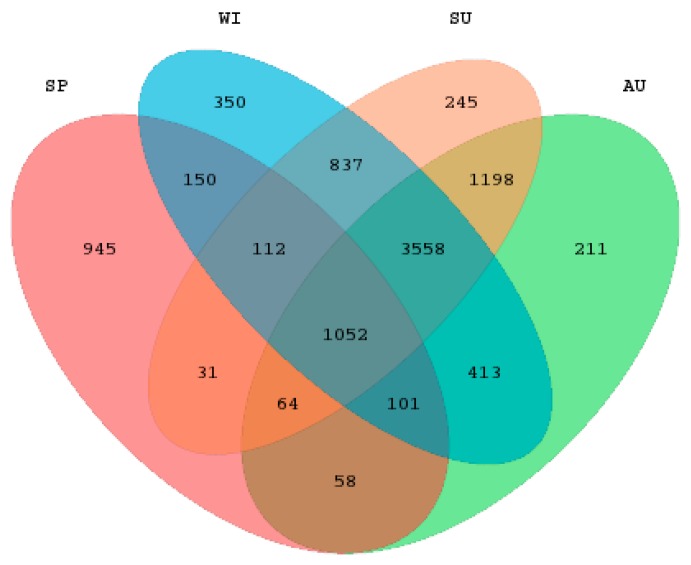
The Venn diagram of operational taxonomic units (OTUs) of sediment samples in spring, summer, autumn, and winter.

**Figure 3 ijerph-16-01931-f003:**
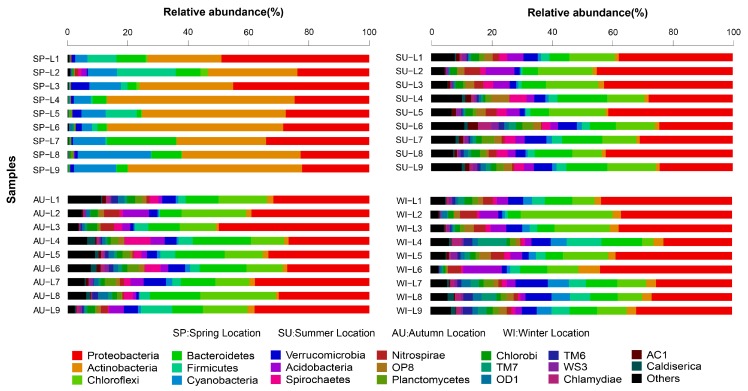
Relative abundance and composition of bacterial phyla detected in sediments of Sancha Lake in the four seasons.

**Figure 4 ijerph-16-01931-f004:**
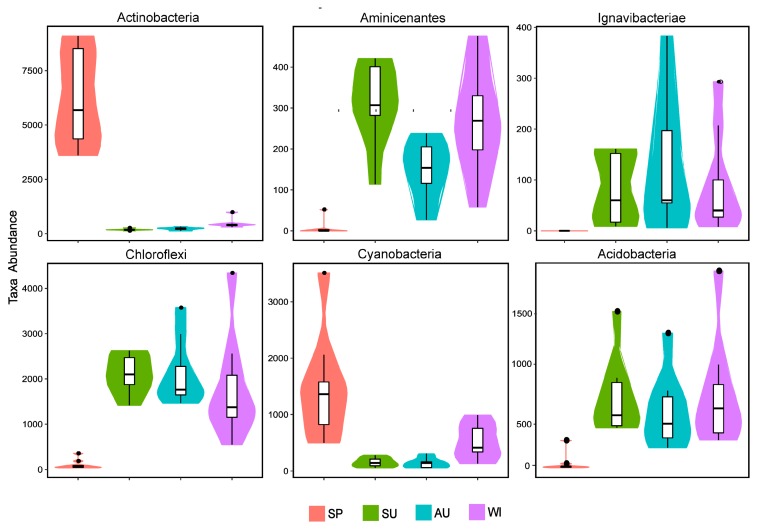
Abundance distributions of the six microbial populations with the most significant differences in the four seasons.

**Figure 5 ijerph-16-01931-f005:**
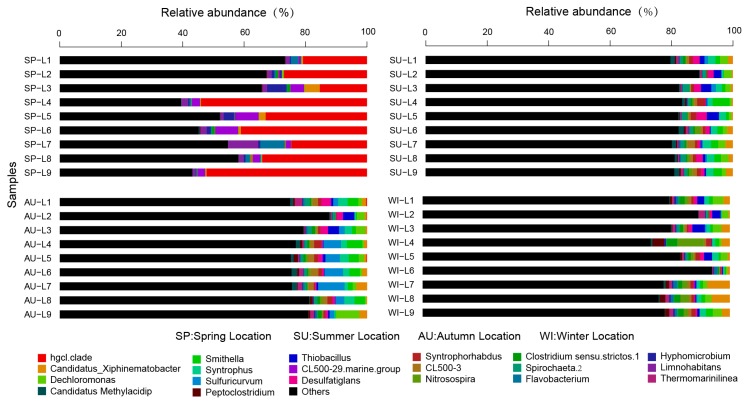
Relative abundance and composition of bacterial genera detected in sediments of Sancha Lake in four seasons.

**Figure 6 ijerph-16-01931-f006:**
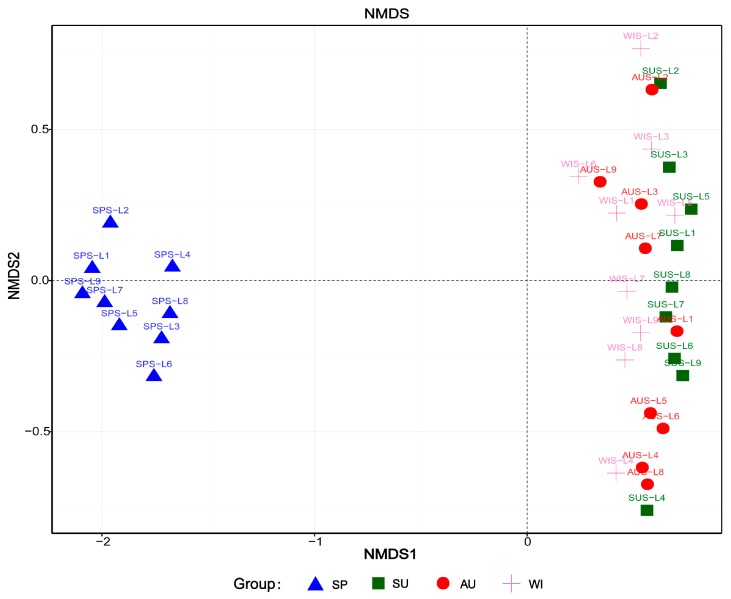
Representation of bacterial community structure based on Weighted UniFrac.

**Figure 7 ijerph-16-01931-f007:**
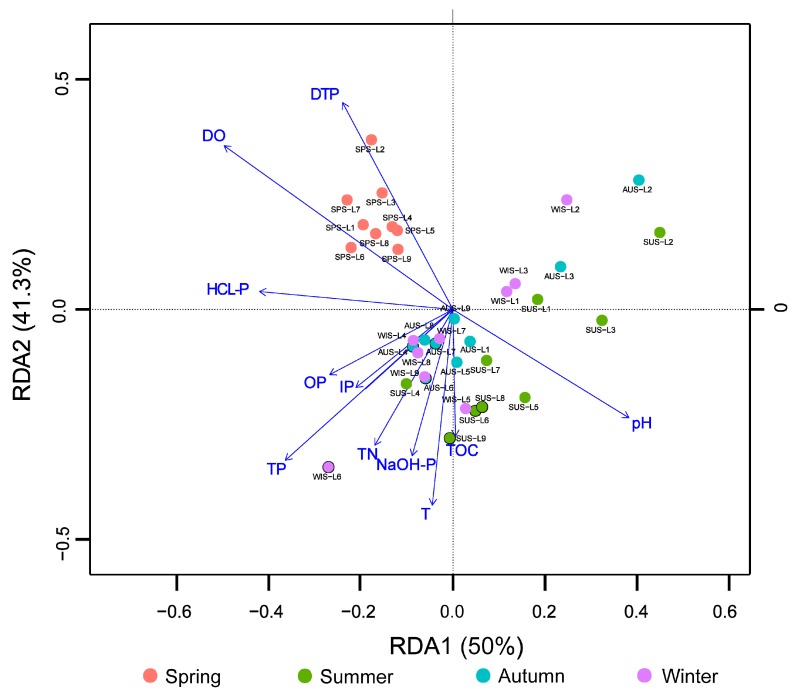
Redundancy analysis (RDA) analysis of bacteria phyla and physico-chemical factors in Sancha Lake.

**Table 1 ijerph-16-01931-t001:** Physicochemical factors of sediments and overlying water in different seasons.

Season	Spring	Summer	Autumn	Winter
pH	6.36 ± 0.17 b	7.49 ± 0.088 a	7.26 ± 0.30 a	6.96 ± 0.17 a
T (°C)	13.16 ± 0.80 c	17.93 ± 1.28 a	15.45 ± 1.31 b	12.10 ± 0.17 c
DO (mg·L^−1^)	8.25 ± 0.82 a	6.27 ± 1.04 b	5.11 ± 0.65 c	5.44 ± 0.74 c
DTP (mg·L^−1^)	0.12 ± 0.150 a	0.016 ± 0.007 b	0.024 ± 0.008 b	0.027 ± 0.012 b
TOC (mg·g^−1^)	21.28 ± 4.55 b	27.03 ± 5.14 a	22.43 ± 3.82 ab	21.45 ± 5.24 b
TN (mg·g^−1^)	1.92 ± 0.42 b	2.56 ± 0.43 a	1.97 ± 0.49 b	1.73 ± 0.63 b
TP (mg·kg^−1^)	1247.60 ± 716.26 b	2266.55 ± 2.04 a	2090.56 ± 1.57 a	1306.93 ± 1.00 b
IP (mg·kg^−1^)	862.37 ± 552.21 b	1116.74 ± 1.30 a	1616.00 ± 1.29 a	991.21 ± 662.21 b
NaOH-P (mg·kg^−1^)	172.46 ± 110.88 b	377.54 ± 239.05 a	378.46 ± 312.10 a	167.21 ± 77.35 b
HCl-P (mg·kg^−1^)	773.25 ± 563.11 b	999.24 ± 1.10 a	1237.74 ± 1.14 a	827.92 ± 1.06 b
OP (mg·kg^−1^)	268.86 ± 121.24 b	314.78 ± 123.09 a	319.61 ± 124.60 a	289.25 ± 164.85 b

Note: pH, temperature (T), dissolved oxygen (DO), and total phosphorus (DTP) were measured in the overlying water of sediments; total organic carbon (TOC), total nitrogen (TN), total phosphate (TP), inorganic phosphorus (IP), organic phosphorus (OP), phosphorous hydrochloride (HCl-P), and phosphonium hydroxide (NaOH-P) were measured in sediments. Data are means ± stand deviation. In the same row, data with different letter such as a, b, and c indicate significant differences, while data with the same letter indicated insignificant differences at 0.05 level. Data with letters ab were insignificantly different from both data with letter a and data with letter b.

**Table 2 ijerph-16-01931-t002:** Correlation coefficients between bacteria groups and physico-chemical factors.

Bacteria Group	pH	T	DO	DTP	TOC	TN	TP	IP	NaOH-P	HCl-P	OP
Proteobacteria	0.28	−0.026	−0.27	−0.226	0.158	0.178	−0.273	−0.23	−0.322	−0.234	−0.25
Actinobacteria	−0.761 **	−0.212	0.639 **	0.540 **	−0.127	−0.071	−0.344 *	−0.089	−0.194	−0.341 *	−0.048
Bacteroidetes	−0.052	−0.284	0.362	0.32	−0.068	0.053	0.106	0.108	0.005	0.203	0.064
Firmicutes	−0.052	−0.284	0.362	0.32	−0.068	0.053	0.106	0.108	0.005	0.203	0.064
Cyanobacteria	−0.674 **	−0.226	0.544 **	0.33 *	−0.321	−0.177	−0.281 *	−0.114	−0.277	−0.102	−0.330 *
Verrucomicrobia	0.296	0.021	−0.362	−0.117	−0.195	0.260	0.218	0.157	0.259	0.103	−0.101
Chloroflexi	0.685 **	0.277	−0.555	−0.437 **	0.239	0.549 **	−0.043	−0.092	0.071	−0.139	−0.086
Acidobacteria	0.463 **	0.179	0.544 **	−0.349 *	0.165	0.311	−0.091	−0.051	−0.06	−0.095	−0.077
Planctomycetes	0.474 **	0.293	−0.438 **	−0.169	0.074	0.044	0.330 *	0.229	0.159	0.408 *	0.124
Spirochaetae	0.243	0.221	−0.236	−0.2047	−0.06	−0.319	−0.153	−0.302	0.243	0.201	0.104
Nitrospirae	0.330 **	0.085	−0.233	−0.322	0.145	0.056 *	−0.129	−0.307	−0.288	−0.048	0.026
Aminicenantes	0.538 **	0.234	−0.466 **	−0.397 *	0.142	0.265	0.232	0.105	0.275	0.09	0.029
Ignavibacteriae	0.635 **	0.261	−0.438 **	−4.3 **	0.277	0.10	−0.188	−0.221	−0.036	−0.236	−0.125

Note: *: Significance at 0.05 level; **: Significance at 0.01 level.

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
