# Peer review of "Microbial Community Structure in the Sediments and Its Relation to Environmental Factors in Eutrophicated Sancha Lake"

_ijerph, 2019, doi:10.3390/ijerph16111931_

Round 1

Reviewer 1 Report

References are not prepared according IJERPH rules: authors should pay an attention on the journal demands, i.e. journal abbreviations should be applied, journal names should be written with italic style, the year should be bolded etc.

Abstract is too long. According IJERPH demands the abstract should be a total of about 200 words maximum. However, the current version of abstract include 361 words, so this section have to be shortened. In the abstract section only the most important and general conclusions from the study should be presented.

Line 19 – authors should precise which 11 environmental factor they studied

Introduction section should be supplemented on information about identified microorganisms in the lakes and sediments. Authors wrote that: (line 58) “there are few studies on the diversity and structure of sedimentary microbial communities in eutrophic sub-deep reservoirs” – here they should shortly present the current knowledge in the context of this statement.  

Lines 129, 131, 194 – should be OTU instead of OUT

Line 165 – should be: the highest level was achieved in spring, ...and the lowest...

Line 167 – should be: with the highest values in spring...and the lowest...

Line 220 – should be Chao1

Line 349 – should be: Winters et al. [24]

Line 351 –should be Jing et al. [25]

Line 352 – should be Yang et al. [7]

Line 359 – should be Yandigieri et al. [26] – please pay an attention that you have 26 citations only...why you wrote [27]???

Line 366 – should be: Li et al. [9]

Discussion is poorly written, sections from 3.1 to 3.4. are not discussed at all.

In my opinion Discussion should be separated from Results and prepared more carefully with more details and more the newest papers should be cited.

Conclusions also demands to be rewritten. The system with pointing is not accurate. Authors should rather clearly summarize they founding’s.

Author Response

Dear Editor,

Thank you for reviewing our manuscript and giving us valuable suggestions to introverts quality. The authors have carefully studied the comments by the reviewers. We thank the reviewers for the thoughtful comments leading to the improvement of our manuscript. All the changes in the updated manuscript are highlighted in Red. The following is the response to the reviewers’ comments.

Comments and Suggestions for Authors

1.References are not prepared according IJERPH rules: authors should pay an attention on the journal demands, i.e. journal abbreviations should be applied, journal names should be written with italic style, the year should be bolded etc.

Reply: .References have been modified according to IJERPH rules. Other references have been added to the text.

2.Abstract is too long. According IJERPH demands the abstract should be a total of about 200 words maximum. However, the current version of abstract include 361 words, so this section have to be shortened. In the abstract section only the most important and general conclusions from the study should be presented.

Reply: Abstract has been modified and only the most important and general conclusions were presented.

3. Line 19 – authors should precise which 11 environmental factor they studied

Reply: Only some of 11 related environmental factors were list for limited words of abstract.

“11 related environmental factors such as pH, TP, TN and DTP were investigated.”

4 Introduction section should be supplemented on information about identified microorganisms in the lakes and sediments. Authors wrote that: (line 58) “there are few studies on the diversity and structure of sedimentary microbial communities in eutrophic sub-deep reservoirs” – here they should shortly present the current knowledge in the context of this statement.  

Reply: Introduction section has been supplemented on information about identified microorganisms in the sediments and present he current knowledge about the diversity and structure of sedimentary microbial communities in eutrophic sub-deep reservoirs.

5. Lines 129, 131, 194 – should be OTU instead of OUT. Line 165 – should be: the highest level was achieved in spring, ...and the lowest.... Line 167 – should be: with the highest values in spring...and the lowest... Line 220 – should be Chao1.Line 349 – should be: Winters et al. [24].Line 351 –should be Jing et al. [25].Line 352 – should be Yang et al. [7].Line 359 – should be Yandigieri et al. [26] – please pay an attention that you have 26 citations only...why you wrote [27]???Line 366 – should be: Li et al. [9]

Reply: Agreed and revision is done.

6.Discussion is poorly written, sections from 3.1 to 3.4. are not discussed at all.

Reply: Agreed and revision is done.

7.Conclusions also demands to be rewritten. The system with pointing is not accurate. Authors should rather clearly summarize they founding’s.

Reply: Agreed and revision is done.

Reviewer 2 Report

General comments

The title seems to point the reader to think that the work is entirely on how the microbial diversity and structure changes with respect to physicochemical parameters. However, the introduction is entirely on eutrophication and Total phosphorus. I therefore suggest that the title be modified to reflect the eutrophication. Also, this topic is very similar to the one previously published by the authors (“Gcd Gene Diversity of Quinoprotein Glucose Dehydrogenase in the Sediment of Sancha Lake and Its Response to the Environment” Int. J. Environ. Res. Public Health 2019, 16, 1).

Line 162 -184: The authors seem to focus on the results and do not discuss the observations. What accounted for the seasonal changes in the parameters? What human activities occurred around L9 that made it have the highest TP? It is not enough to just say “human activities”. What cage culture practice influenced the observations at L8, L6, and L5? The same applies to the other sections of the discussion. Another issue is that the authors do not seem to compare their results with any previous studies. There are works that have been conducted on the impact of land uses on the microbial diversity, structure and function in riverbed sediments in many parts of the world. The authors may need to check that to show how similar to or different from other works their work is.

Also, how statistically different were the values between the seasons (the authors performed ANOVA).

These results are linked to eutrophication and therefore should reflect some form of pollution. Which of the values were within the country’s acceptable limits (and what are those limits?) and which ones were not? E.g. At Line 180, the authors say, “Environmental Protection Agency (EPA) standard.” What are those standard values and please include a reference for that.

Line 286 – 303: It is evident from the authors discussions that the seasons played a role in microbial abundance. The authors indicate that the spring season had greater abundance compared to non-spring seasons. What explains this? What are the environmental changes that occur during that season that favours the microbial abundance? The same holds for Lines 306 – 310 and section 3.5.

Throughout the manuscript, the authors use sediment and sediments interchangeably. I think they need to maintain “sediments” throughout, except when referring to “sediment sample(s)”

Specific comments

Line 18: regions of the 16S rRNA gene,

Line 23: The results demonstrated four main patterns. (1)…….

Line 26: The seasonal variation in the diversity of microbial populations in Sancha Lake sediment was significant

Line 27 - 28: The microbial populations in the four seasons belonged to two……

Line 29: Delete “eutrophic”

Line 32 – 34: The abundance of Actinobacteria and Cyanobacteria was strongly and significantly correlated with the dissolved total phosphorus (DTP) of the overlying water.

Line 35: highly diverse

Line 61: structure of the microbial

Line 61 – 67: Please split into shorter sentences. This sentence is too long and complex making it difficult to understand.

Lines 87 – 99: I think that the same authors have previously reported the same procedure and should therefore just refence this rather than trying to rewrite the procedures entirely. Please see “Gcd Gene Diversity of Quinoprotein Glucose Dehydrogenase in the Sediment of Sancha Lake and Its Response to the Environment” Int. J. Environ. Res. Public Health 2019, 16, 1

Line 104: the 16S rRNA gene

Line 118: The purified amplicons were…

Line 123: control of original sequences

Line 131: OUT, R was used for (Please change all instances of “the R” to just “R”)

Line 142: Delete “of”

Line 154: Please write SPSS in full on first instance

Line 153 – 159: At what alpha were your results considered statistically significant?

Line 163: 6.06~7.73, and was weakly…..

Line 164: of the overlying water

Line 168 – 169: “and was close in autumn and winter” what does this mean? Also check line 170

Line 172: Delete “regarding the same sampling site”

Line 176: relatively…

Line 190 – 191: “By with on the 16S rRNA gene of the 190 sediment DNA,” Please rephrase this. I suggest “High-throughput sequencing of the 16S rRNA was performed on an Illumina Miseq…..

Line 194: OUT

Figure 2 and 3: What do the bars on the curves represent? Please indicate this as a note under each figure.

Line 248: 64 phyla, while the site L9

Line 262: it was significantly different

Line 263 – 265: “The distribution of the microbial populations was with the most species in summer, then winter and autumn, and the least in spring.” What do the authors mean by this? The sentence needs to be rephrased.

Line 265: Do the authors mean “The most abundant microbial populations”?

Line 277: Do the authors mean that “eutrophication forms Actinobacteria and Cyanobacteria”? Please rephrase.

Line 314 – 315: …environmental physicochemical factors of the dominant microbial phyla”? Where the environmental factors for the water and sediments or for the dominant microbial phyla?

Line 321: that they had…

Line 322: and the DTP changes were correlated with microbial communities extremely significantly in spring. Please rephrase. The “extremely significantly” expression is not clear. Same with line 329.

Line 370 – 381: Please write this as a single paragraph.

Author Response

Dear Editor,

Thank you for reviewing our manuscript and giving us valuable suggestions to introverts quality. The authors have carefully studied the comments by the reviewers. We thank the reviewers for the thoughtful comments leading to the improvement of our manuscript. All the changes in the updated manuscript are highlighted in Red. The following is the response to the reviewers’ comments.

General comments

1. The title seems to point the reader to think that the work is entirely on how the microbial diversity and structure changes with respect to physicochemical parameters. However, the introduction is entirely on eutrophication and Total phosphorus. I therefore suggest that the title be modified to reflect the eutrophication. Also, this topic is very similar to the one previously published by the authors (“Gcd Gene Diversity of Quinoprotein Glucose Dehydrogenase in the Sediment of Sancha Lake and Its Response to the Environment” Int. J. Environ. Res. Public Health 2019, 16, 1).

 Reply: Agreed and revision is done.

The title has been modified:  Microbial Community Structure in the Sediments and its relation with Environmental Factors in Eutrophicated  Sancha Lake

The introduction has been supplemented.

2.Line 162 -184: The authors seem to focus on the results and do not discuss the observations. What accounted for the seasonal changes in the parameters? What human activities occurred around L9 that made it have the highest TP? It is not enough to just say “human activities”. What cage culture practice influenced the observations at L8, L6, and L5? The same applies to the other sections of the discussion. Another issue is that the authors do not seem to compare their results with any previous studies. There are works that have been conducted on the impact of land uses on the microbial diversity, structure and function in riverbed sediments in many parts of the world. The authors may need to check that to show how similar to or different from other works their work is.

 Reply: Agreed and revision is done.

Discussion has been supplemented in sections of 3.1 - 3.4 and we compared our results with those of other previous studies.

Human activities have been detailed: The sampling sites L8, L6, L5, and L9 had high sediments TP contents, with L8 and L9 the highest. Historically, those sampling sites were the high-concentration areas for cage culture, thus the high TP contents may be caused by excessive fish feed accumulated on the lake bottom. The site L9 is adjacent to high human activity area, and the high TP contents may be due to the deposition of OM discharged from the wastewater.

3.Also, how statistically different were the values between the seasons (the authors performed ANOVA).

Reply: Agreed and revision is done.

Statistical significance was determined at the confidence levels of 0.05.

4.These results are linked to eutrophication and therefore should reflect some form of pollution. Which of the values were within the country’s acceptable limits (and what are those limits?) and which ones were not? E.g. At Line 180, the authors say, “Environmental Protection Agency (EPA) standard.” What are those standard values and please include a reference for that.

  Reply: Agreed and revision is done.

There are no the country’s acceptable pollutant limits of lake sediments. So we refer to the “Nutrient Criteria Technical Guidance Manual for Lakes and Reservoirs” developed by US EPA. The sediment TP levels suggested heavily polluted (>650 mg•kg-1).

5.Line 286 – 303: It is evident from the authors discussions that the seasons played a role in microbial abundance. The authors indicate that the spring season had greater abundance compared to non-spring seasons. What explains this? What are the environmental changes that occur during that season that favours the microbial abundance? The same holds for Lines 306 – 310 and section 3.5.

  Reply: In spring, microbial community structure in the sediments of Sancha lake was different from that non-spring seasons for DO increasing and water disturbance caused by irrigation activity.

6.Throughout the manuscript, the authors use sediment and sediments interchangeably. I think they need to maintain “sediments” throughout, except when referring to “sediment sample(s)”

   Reply: Agreed and revision is done.

Specific comments 

1.Line 18: regions of the 16S rRNA gene,

Reply: Agreed and revision is done.

2.Line 23: The results demonstrated four main patterns. (1)…….

Reply: Accepted and it is changed.

3.Line 26: The seasonal variation in the diversity of microbial populations in Sancha Lake sediment was significant

Reply: Accepted and it is changed.

4.Line 27 - 28: The microbial populations in the four seasons belonged to two……

Reply: Accepted and it is changed.

5.Line 29: Delete “eutrophic”

Reply: Accepted and it is changed.

6.Line 32 – 34: The abundance of Actinobacteria and Cyanobacteria was strongly and significantly correlated with the dissolved total phosphorus (DTP) of the overlying water.

Reply: Accepted and it is changed.

7.Line 35: highly diverse

Reply: Accepted and it is changed.

8.Line 61: structure of the microbial

Reply: Accepted and it is changed.

9.Line 61 – 67: Please split into shorter sentences. This sentence is too long and complex making it difficult to understand.

 Reply:  Accepted and it is changed.

10.Lines 87 – 99: I think that the same authors have previously reported the same procedure and should therefore just refence this rather than trying to rewrite the procedures entirely. Please see “Gcd Gene Diversity of Quinoprotein Glucose Dehydrogenase in the Sediment of Sancha Lake and Its Response to the Environment” Int. J. Environ. Res. Public Health 2019, 16, 1

 Reply: Accepted and it is changed.

11.Line 104: the 16S rRNA gene

Reply: Accepted and it is changed.”

12.Line 118: The purified amplicons were…

Reply: Accepted and it is changed.

13.Line 123: control of original sequences

Reply: Accepted and it is changed.

14.Line 131: OUT, R was used for (Please change all instances of “the R” to just “R”

Reply: Accepted and it is changed.

15.Line 142: Delete “of”

Reply: Accepted and it is changed.

16.Line 154: Please write SPSS in full on first instance

Reply: statistical product and service solutions (SPSS)

17.Line 153 – 159: At what alpha were your results considered statistically significant?

Reply: Statistical significance was determined at the confidence levels of 0.05

18.Line 163: 6.06~7.73, and was weakly…..

Reply: Accepted and it is changed.

19.Line 164: of the overlying water

Reply: Accepted and it is changed.

20.Line 168 – 169: “and was close in autumn and winter” what does this mean? Also check line 170

Reply:“close” has been changed into“neutral”

21.Line 172: Delete “regarding the same sampling site”

Reply: Accepted and it is changed.

22.Line 176: relatively…

Reply: Accepted and it is changed.

23.Line 190 – 191: “By with on the 16S rRNA gene of the 190 sediment DNA,” Please rephrase this. I suggest “High-throughput sequencing of the 16S rRNA was performed on an Illumina Miseq…..

Reply: Accepted and it is changed.

24.Line 194:

Reply: Accepted and it is changed.

25.Figure 2 and 3: What do the bars on the curves represent? Please indicate this as a note under each figure.

Reply: Figure 2 and 3 have been deleted for another expert’s suggestion.

26.Line 248: 64 phyla, while the site L9

Reply: Accepted and it is changed.

27.Line 262: it was significantly different

Reply: Accepted and it is changed.

28.Line 263 – 265: “The distribution of the microbial populations was with the most species in summer, then winter and autumn, and the least in spring.” What do the authors mean by this? The sentence needs to be rephrased.

 Reply: The sentence “The distribution of the microbial populations was with the most species in summer, then winter and autumn, and the least in spring.” has been rephrased: “Microbial species in the sediments was the most abundant in summer and was the least abundant in spring.”

29.Line 265: Do the authors mean “The most abundant microbial populations”?

Reply:“The dominant microbial populations”has been changed into“The microbial populations (relative species abundance >1%)”.

30.Line 277: Do the authors mean that “eutrophication forms Actinobacteria and Cyanobacteria”? Please rephrase.

Reply: I mean Actinobacteria and Cyanobacteria might be the key populations or indicator organisms in eutrophication. Therefore the sentence is rephrased: Therefore, they might be the key populations or indicator organisms in eutrophication.

31.Line 314 – 315: …environmental physicochemical factors of the dominant microbial phyla”? Where the environmental factors for the water and sediments or for the dominant microbial phyla?

Reply: I mean environmental physicochemical factors for the water and sediments. And “The RDA was performed on the relative abundance and the environmental physicochemical factors of the dominant microbial phyla â€ťhas been rephrased:“The RDA was performed on the relative abundance of the microbial phyla(relative species abundance >1%) and the environmental physicochemical factors for the water and sediments”

32.Line 321: that they had…

 Reply: Accepted and it is changed.

33.Line 322: and the DTP changes were correlated with microbial communities extremely significantly in spring. Please rephrase. The “extremely significantly” expression is not clear. Same with line 329.

 Reply: “extremely significantly” has been changed into:  â€śhighly significantly”

34.Line 370 – 381: Please write this as a single paragraph.

Reply: Agreed and revision is done.

Reviewer 3 Report

The submitted article looks like an unfinished investigation. The introduction is very weak, short. the authors placed in the article figures based on the number of operational taxonomic units and number of sequences which should not be not shown - Fig.2 and 3, table 2 (technical data with no scientific value). Fig5 - too much data, not clear. The purpose of the work has not been clearly stated.

Author Response

Dear Editor,

Thank you for reviewing our manuscript and giving us valuable suggestions to introverts quality. The authors have carefully studied the comments by the reviewers. We thank the reviewers for the thoughtful comments leading to the improvement of our manuscript. All the changes in the updated manuscript are highlighted in Red. The following is the response to the reviewers’ comments.

Comments and Suggestions for Authors

1. The submitted article looks like an unfinished investigation. The introduction is very weak, short. the authors placed in the article figures based on the number of operational taxonomic units and number of sequences which should not be not shown - Fig.2 and 3, table 2 (technical data with no scientific value). Fig5 - too much data, not clear. The purpose of the work has not been clearly stated.

Reply: Agreed and revision is done.

Introduction has been modified and supplemented. Fig.2 and 3 have been deleted. Fig.5 has been modified.

Round 2

Reviewer 1 Report

-

Author Response

Thanks for the expert opinion.

Reviewer 2 Report

Please go through the entire manuscript again to amend minor grammatical errors.

Line 34: and have…

Line 35: sediments

Line 36: sediments (please check all other instances and correct accordingly)

Line 107: were measured as previously described by Li et al. [14].

Line 185: The site L9 is adjacent to a high human activity area

Line 199 – 201: pH, T, DO and DTP were measured….. HCl-P and NaOH-P were measured …..

Please define a, b, c and ab

Line 252: phyla

Line 274: sediments were the most abundant in summer and the least abundant

Line 326: DO. Therefore,

Line 347 – 348: and the DTP changes were highly significantly correlated with microbial communities in spring.

Author Response

1. Line 34: and have…

Reply: Accepted and it is changed.

2. Line 35: sediments

 Reply: Accepted and it is changed.

3.Line 36: sediments (please check all other instances and correct accordingly)

 Reply: Checked and it is changed.

4.Line 107: were measured as previously described by Li et al. [14].

 Reply: Accepted and it is changed.

5.Line 185: The site L9 is adjacent to a high human activity area

   Reply: Accepted and it is changed.

6.Line 199 – 201: pH, T, DO and DTP were measured….. HCl-P and NaOH-P were measured …..

 Reply: Accepted and it is changed.

7. Please define a, b, c and ab

Reply: Accepted and it is done: Data were means±stand deviation. In the same row, data with different letter such as a, b and c indicated significant difference while data with the same letter indicated insignificant difference at 0.05 level. Data with letters ab were insignificantly different from both data with letter a and data with letter b.

 8.Line 252: phyla

  Reply: Accepted and it is changed.

9.Line 274: sediments were the most abundant in summer and the least abundant

  Reply: Accepted and it is changed.

10.Line 326: DO. Therefore,

  Reply: Accepted and it is changed.

11.Line 347 – 348: and the DTP changes were highly significantly correlated with microbial communities in spring.

 Reply: Accepted and it is changed.

Reviewer 3 Report

The authors have made some improvements. However, the text is still not ready for publication. the presented work can be a good beginning of the further investigations.

Table 2 with techical data (number of sequences - without scientific value) is still present. fig 3 is not described well - are there data from various seasons? If yes, it should be written. I don't understand the reason for placing in the text Fig. 6.

The results presented deal  with phyla, there are no more detailed information on family or genus level what would be more valuable.  The conclusions are still deficient.

Author Response

Dear Editors and Reviewer:

Revised portion are marked using the "Highlight" function in the paper. The main corrections in the paper and the responds to the reviewer’s comments are as flowing:

Responds to the reviewer’s comments:

1.     Table 2 with techical data (number of sequences - without scientific value) is still present.

Reply: Deleted the number of sequences in Table 2

2.     fig 3 is not described well - are there data from various seasons? If yes, it should be written.

Reply: Fig.3 has been modified. there were data from various seasons, it was written.

3. I don't understand the reason for placing in the text Fig. 6. 

Reply: Fig. 6 and the relative methods and reference have been deleted

3.     The results presented deal  with phyla, there are no more detailed information on family or genus level what would be more valuable. 

Reply: the information on family and genus level has been supplemented.

 4.The conclusions are still deficient.

The conclusions have been modified.

Special thanks to you for your good comments

Round 3

Reviewer 3 Report

The authors have added the results of analysis of microorganism abundance at the genus level  what I do appreciate. 

table 2 -technical data, not needed in the text

Fig. 6 - a comparison with the results of other groups' investigation is needed

Author Response

Dear Editors and Reviewers:

Revised portion are marked using the "Track Changes" function in the paper. The main corrections in the paper and the responds to the reviewer’s comments are as flowing:

Responds to the reviewer’s comments:

1. table 2 -technical data, not needed in the text

Reply: Accepted and deleted the Table 2.

2.Fig. 6 - a comparison with the results of other groups' investigation is needed

Reply: Accepted and it was changed.